# Personalized Medicine of Omega-3 Fatty Acids in Depression Treatment in Obese and Metabolically Dysregulated Patients

**DOI:** 10.3390/jpm13061003

**Published:** 2023-06-15

**Authors:** Suet-Kei Wu, Wei-Jen Chen, Jane Pei-Chen Chang, Ta-Wei Guu, Ming-Che Hsin, Chih-Kun Huang, David Mischoulon, Lucile Capuron, Kuan-Pin Su

**Affiliations:** 1Graduate Institute of Nutrition, China Medical University, Taichung 404, Taiwan; wskei23@gmail.com; 2Mind-Body Interface Research Center (MBI-Lab), China Medical University Hospital, Taichung 404, Taiwan; peko80@gmail.com (J.P.-C.C.); da20vid@gmail.com (T.-W.G.); 3An-Nan Hospital, China Medical University, Tainan 709, Taiwan; yizingch@gmail.com; 4Department of Psychiatry, China Medical University Hospital, Taichung 404, Taiwan; 5College of Medicine, China Medical University, Taichung 404, Taiwan; 6Division of Psychiatry, Department of Internal Medicine, China Medical University Beigang Hospital, Yunlin 651, Taiwan; 7Body Science & Metabolic Disorders International Medical Centre (BMIMC), China Medical University & Hospital, Taichung 404, Taiwan; matthsin@gmail.com (M.-C.H.); dr.ckhuang@hotmail.com (C.-K.H.); 8Depression Clinical and Research Program, Massachusetts General Hospital, Harvard Medical School, Boston, MA 02114, USA; dmischoulon@partners.org; 9NutriNeuro, University of Bordeaux, INRAE, Bordeaux INP, UMR 1286, F-33076 Bordeaux, France; lucile.capuron@inrae.fr

**Keywords:** omega-3 polyunsaturated fatty acids (n-3 PUFAs), major depressive disorder (MDD), obesity, metabolic dysregulation, comorbidities

## Abstract

The co-occurrence of depression and obesity has become a significant public health concern worldwide. Recent studies have shown that metabolic dysfunction, which is commonly observed in obese individuals and is characterized by inflammation, insulin resistance, leptin resistance, and hypertension, is a critical risk factor for depression. This dysfunction may induce structural and functional changes in the brain, ultimately contributing to depression’s development. Given that obesity and depression mutually increase each other’s risk of development by 50–60%, there is a need for effective interventions that address both conditions. The comorbidity of depression with obesity and metabolic dysregulation is thought to be related to chronic low-grade inflammation, characterized by increased circulating levels of pro-inflammatory cytokines and C-reactive protein (CRP). As pharmacotherapy fails in at least 30–40% of cases to adequately treat major depressive disorder, a nutritional approach is emerging as a promising alternative. Omega-3 polyunsaturated fatty acids (n-3 PUFAs) are a promising dietary intervention that can reduce inflammatory biomarkers, particularly in patients with high levels of inflammation, including pregnant women with gestational diabetes, patients with type 2 diabetes mellitus, and overweight individuals with major depressive disorder. Further efforts directed at implementing these strategies in clinical practice could contribute to improved outcomes in patients with depression, comorbid obesity, and/or metabolic dysregulation.

## 1. Introduction

Depression and obesity have emerged as significant public health concerns due to their increasing prevalence. Before 2020, depression impacted about 280 million people globally [1], but the COVID-19 pandemic has worsened the situation, leading to an increase of about 53.2 million new cases of major depressive disorder (MDD) and an overall prevalence of 3152.9 cases per 100,000 population [2]. MDD can affect individuals of any age or ethnicity, causing severe individual and socioeconomic consequences [3,4]. Women have a higher risk of developing MDD compared to men [4,5], and adolescents, especially girls and young women, experience a significant increase in the prevalence of depression, self-harm, suicide, and overall [6]. MDD not only impacts mental health but also has significant effects on physical health, functional abilities, workplace productivity, and economic burden [7,8,9]. The economic burden includes the costs of medical treatments, which may not always effectively manage depression or its complications [8,10].

Depression is notorious for being a risk factor for metabolic dysfunction associated with obesity, including an inflammatory response, high blood pressure, impaired insulin sensitivity, and impaired leptin sensitivity [11]. This association suggests a bidirectional relationship between depression and metabolic dysfunction, with each condition increasing the progressive risk of the other by 50–60% [12]. Metabolic abnormalities related to inflammation, commonly observed in individuals with obesity, may play a vital neurobiological role in the onset of depression beyond the influence of obesity alone [13,14,15]. Obese adults have a 14–34% higher risk of clinically diagnosed MDD compared to non-obese controls, while obese adolescents have a 40% increased risk of developing depression [16,17,18]. The combined impact of MDD and obesity on quality of life is greater than their independent effects, suggesting a synergistic relationship between these conditions [19].

Current pharmacotherapy for MDD is often inadequate due to the heterogeneity of patients under the existing diagnostic system. As a result, there is a growing interest in nutritional approaches, such as omega-3 polyunsaturated fatty acid (n-3 PUFA) supplementation, as a potential alternative for treating MDD. A recent practice guideline published by Guu et al. (2019) recommends considering n-3 PUFAs as a possible therapeutic option for MDD based on their beneficial effects on inflammation and mood and on lowering the likelihood of developing MDD in specific populations, including women with perinatal MDD, children, and the elderly [20,21,22,23,24,25]. Recent studies have also investigated the effect of n-3 PUFAs on individuals with high inflammation, including MDD patients [26,27], patients with MDD and comorbid type 2 diabetes mellitus [24,28], or those who are overweight/obese [26].

## 2. MDD, Stress, and Neuropsychoimmunity

MDD is a leading cause of global disability [29] and is diagnosed according to the Diagnostic and Statistical Manual of Mental Disorders (DSM-5) criteria, with symptoms that may vary between individuals. One pathophysiological mechanism of MDD is attributed to the neuroendocrine theory of abnormal homeostasis of the hypothalamus-pituitary-adrenal (HPA) axis, characterized by excessive activation of the stress hormone cortisol [30]. Acute and severe forms of MDD have been associated with an increased cortisol response to stress, whereas mild or atypical types of MDD may not display the same cortisol response [31].

In addition to cortisol, adipokines such as leptin and adiponectin have also been proposed to play significant roles in MDD. Leptin can activate PI3K/Akt and MAPK/Akt pathways that inhibit GSK3β activities and elevate β-catenin signaling and the Akt signaling pathway [32], besides activating hypothalamic brain-derived neurotrophic factor (BDNF)-expressing neurons to induce neuroprotection [33]. Leptin levels are lower in male patients with MDD but higher in those with a BMI ≥ 25 and aged 40 and above [34]. Adiponectin has receptors in the brain and can cross the blood-brain barrier. Its pathways promote neurogenesis [35], spinogenesis, dendritic complexity [36], and synaptic plasticity modulation [37], suggesting it could result in antidepressant responses [38,39].

The neuroimmune hypothesis posits that inflammation may have a role in MDD as evidenced by elevated levels of markers of inflammation and oxidative stress, including interleukin-6 (IL-6), interleukin-17 (IL-17), interleukin-21 (IL-21), interleukin-35 (IL-35), tumor necrosis factor-alpha (TNF-α), 8-hydroxy-2′-deoxyguanosine (8-OHdG), and F2-isoprostanes in individuals with MDD [40,41]. Clinical findings indicate that cytokine administration leads to the development of MDD in 40–50% of patients [42,43,44,45]. Furthermore, chronic neuroinflammation, specifically interleukin-1β (IL-1β) and interferon-α (IFNα), can result in structural, connectivity, and excitability changes in the brain [46]. Notably, chronic systemic inflammation has been identified as a major contributor to neuropsychiatric symptoms in obese individuals, irrespective of depressive history [47].

The classical antidepressant hypothesis, known as the monoamine theory, uses monoamine oxidase inhibitors (MAOIs) and selective serotonin reuptake inhibitors (SSRIs) to restore low monoamine levels in the brain [48], while serotonin and norepinephrine reuptake inhibitors (SNRIs) are also used to treat MDD [49], but individual response varies significantly. Long-term follow-up studies have identified a higher risk of weight gain in individuals prescribed antidepressants, highlighting the need for personalized treatment approaches that consider an individual’s BMI or adiposity [50]. Studies have shown that different medications may be more effective for individuals with different levels of obesity, with escitalopram being more effective for individuals with a normal BMI, venlafaxine for those with obesity classes II and III, and bupropion-SSRIs for those with obesity class III (BMI ≥ 40 kg/m^2^) [51,52]. However, morbid obesity (BMI ≥ 40 kg/m^2^) may complicate drug response outcomes [53], suggesting the need for personalized treatment approaches.

Despite advances in MDD treatment, approximately 30% failed to achieve adequate clinical response [54], while higher BMI has been associated with poorer treatment outcomes [55,56] and increased side effects leading to lower compliance [57,58]. These individual factors, including BMI and adiposity, highlight the need to improve tolerability and efficacy, with an understanding of the complex interplay between neuroendocrine, adipokine, and neuroimmune factors and the role of BMI in treatment response being crucial for advancing personalized therapeutic approaches in clinical practice.

## 3. The Interplay between Depression, Obesity, and Metabolic Dysregulation

Obesity, characterized by excessive accumulation of adipose tissue in the body and affecting over one billion people worldwide, is a significant global health issue associated with an increased risk of ischemic heart disease, type 2 diabetes, and certain cancer cases [59]. The body mass index (BMI) is commonly used to diagnose obesity. However, the application of international BMI criteria to Asian and Pacific populations has been controversial, leading to the development of regional criteria. This underscores the importance of considering population-specific factors, as body composition and adipose tissue distribution can vary across different ethnicities.

Obesity-induced inflammation, characterized by inflammatory processes in adipose tissue, has been linked to mood symptoms associated with obesity [60,61,62]. Chronic inflammation is caused by enlarged adipocytes and disruption of adipose tissue function, triggering local and systemic inflammation [63], which is closely associated with metabolic syndrome and damage to various organ systems [64]. Furthermore, immune challenges can activate adipose tissue hypertrophy, inflammation, macrophage infiltration, and secretion of pro-inflammatory cytokines such as leptin, interleukin-6 (IL-6), tumor necrosis factor (TNF-α), and plasminogen activator inhibitor-1 (PAI-1) [65,66,67], leading to inflammation in the brain and potentially triggering depression [46].

Obesity and MDD share common mechanisms such as dysregulation of glucose and lipid metabolism, persistent inflammation, and excessive stimulation of the HPA axis [17,68]. Cortisol has been shown to increase appetite and promote the consumption of high-calorie foods, potentially contributing to weight gain and obesity development [17]. Chronic exposure to elevated cortisol levels can lead to structural changes in brain areas, such as the hippocampus, potentially contributing to cognitive deficits commonly seen in patients with MDD [69]. Furthermore, individuals with MDD have been found to exhibit altered cortisol circadian rhythms and blunted cortisol responses to stress, which may be related to HPA axis dysregulation and elevated cortisol levels [69]. Additionally, certain antidepressant medications used to treat MDD may exacerbate obesity through side effects related to weight gain and changes in appetite [70,71].

Leptin resistance is linked to obesity and its associated metabolic imbalances [72]. Adiponectin may have a protective role against the development of metabolic dysregulation [73,74,75]. Adiponectin levels may fluctuate in individuals with MDD and are influenced by factors such as gender [76] and depression severity [77]. Research has also found interleukin-6 (IL-6) increases and expression of adiponectin (ADIPOQ) declines in the adipose tissue of depressed patients compared to non-depressed individuals [78]. The severity of depression and adiponectin levels may be correlated, suggesting possible direct involvement in the pathogenesis of depression or the presence of coexisting metabolic disturbances [79].

In addition, the gut microbiota plays a significant role in regulating various physiological processes that affect both obesity and MDD [80]. An imbalance in the microbial community, known as gut dysbiosis, has been implicated as a significant factor in the development of both conditions. “Leaky gut” is associated with increased intestinal permeability and has been linked to both inflammatory bowel disorder (IBD) and MDD, particularly in patients with lower levels of beneficial bacteria such as Bifidobacterium and Lactobacillus [81,82,83]. The abundance of Bacteroidetes and Firmicutes has been proposed as biomarkers of obesity and linked to IBD [82], and their ratio is significantly elevated in individuals who are overweight or obese [84]. N-3 PUFAs derived from flaxseed [85] and fish oil [86] have been shown to lower their populations. These anaerobic bacteria ferment fiber and produce short-chain fatty acids (SCFA), which are essential metabolites for maintaining intestinal homeostasis, regulating immune function, and having systemic effects on metabolism and cardiovascular health [87]. In addition, elevated intestinal fatty acid binding protein (I-FABP) levels have been observed in patients with recent suicide attempts and are positively correlated with the severity of depressive symptoms [88], while distinct blood microbiome and metabolomic signatures have been observed in depressed patients following antidepressant treatment [89].

## 4. Personalized Omega-3 Polyunsaturated Fatty Acids (n-3 PUFAs) Intervention in Depression

Low levels of n-3 PUFAs, including EPA and DHA, have been observed in depressed patients [90,91], particularly those who do not respond to antidepressants [92]. N-3 PUFAs have been identified as potential therapeutic agents for depression due to their numerous beneficial effects, including anti-neuroinflammatory, anti-oxidative stress, anti-neurodegeneration, and modulation of the neurotransmitter system [93]. Furthermore, supplementation with these fatty acids in the form of fish oil is comparable to commonly used antidepressant drugs [94,95]. Therefore, supplementation of these fatty acids to maintain adequate levels of n-3 PUFAs in the body is crucial.

Mehdi et al. demonstrated the use of omega-3 as an adjunct therapy to reduce the severity of depression [5]. Current recommendations suggest using pure EPA or an EPA-DHA combination as adjunctive therapy for depression at 1–2 g/day for at least 8 weeks [23]. These findings suggest that supplementation with PUFAs may be a potential alternative approach for managing MDD, notably in pregnant women [20,21,22,96], children, and the elderly. A recent meta-analysis demonstrated that n-3 PUFAs have a positive antidepressant effect in MDD patients, with an effect size of 0.4 [97]. It is noteworthy that a greater dosage of at least 60% EPA in the intervention group increases the likelihood of achieving a positive antidepressant response [97]. The effectiveness of EPA supplementation appears to be higher when taken at a dosage of 4 g/day [26,98]. Additionally, Tu et al. investigated the efficacy of EPA and DHA using functional magnetic resonance imaging (fMRI), observing elevated brain activity in emotion perception and cognitive control regions [99]. However, the efficacy of n-3 PUFAs may vary depending on the presence of comorbidities in depressed individuals [100], highlighting the need for personalized treatment strategies tailored to each patient’s unique needs and symptoms.

Individuals’ biological characteristics and safety concerns should be considered when personalizing nutritional approaches for the management of MDD. Depressed patients often have co-existing inflammatory conditions, neurodegenerative diseases, or physiological deficits (see Table 1). N-3 PUFAs have shown effectiveness in improving depressive symptoms, especially in those with inflammation. For instance, depression is a common occurrence in patients with hepatitis C who receive interferon (IFN)-α therapy. However, it was found that EPA, but not DHA, significantly reduced the occurrence of IFN-α depression among patients undergoing the therapy [101]. It also indicates that supplementation of n-3 PUFAs is highly effective in improving depression, with better clinical effects in pure EPA and EPA-major groups compared to DHA-major groups [102], particularly in individuals with high inflammatory status. Omega-3 indices were found to improve cognitive depressive symptoms in depressed patients with chronic heart failure [103]. Furthermore, the effectiveness of n-3 PUFA supplementation in depressed patients with comorbid cardiovascular disease was found to be dependent on depression severity. Specifically, n-3 PUFA supplementation (2 g EPA and 1 g DHA daily) only led to improvement in core depression symptoms among patients with very severe MDD when the treatment was stratified based on depression severity [104].

Neuropsychiatric symptoms, including depressive symptoms, are common in most neurodegenerative diseases, such as Alzheimer’s disease and Parkinson’s disease; however, they may be only partially or not at all responsive to conventional antidepressant therapies [118,119]. N-3 PUFAs have shown pleiotropic effects on neural structure and function and play a critical role in regulating mood in the brain. Although the exact mechanisms are not yet understood, n-3 PUFAs can potentially improve depressive symptoms and slow cognitive decline in patients with neurodegenerative diseases such as Alzheimer’s and Parkinson’s [93]. In a study by Lin et al. (2022), n-3 PUFA treatment using 0.8 g EPA and 0.35 g DHA daily or pure 1.6 g EPA daily in Alzheimer’s patients resulted in a clinical improvement of the inflammatory biomarker chemokine ligand 4 (CCL4) and a slowdown in cognitive decline, specifically in the spoken ability domain [120]. Similarly, Freund-Levi et al. (2008) found that n-3 PUFA treatment with 1.7 g DHA and 0.6 g daily supplementation led to significant improvements in neuropsychiatric symptoms, including potentially positive effects on depressive symptoms in non-APOE omega4 carriers and agitation symptoms in APOE omega4 carriers among patients with mild to moderate Alzheimer’s disease [111]. Chiu et al. (2008) demonstrated that n-3 PUFA monotherapy exhibited a higher level of eicosapentaenoic acid on red blood cell membranes and a significant improvement in the cognitive portion of the Alzheimer’s Disease Assessment Scale (ADAS-cog) compared to the placebo group in participants with mild cognitive impairment. However, this may not be as effective for those with Alzheimer’s disease [110].

Additionally, da Silva et al. (2008) investigated the effect of chronic supplementation with fish oil containing 1.2 g/day of n-3 PUFAs and found that it led to a high rate of remission of depressive symptoms over 3 months, regardless of whether the Parkinson’s patients were taking antidepressant medication or not [109]. Moreover, a recent study by Borsini et al. (2021) demonstrated the neuroprotective effects of EPA and DHA treatment in preventing neurodegeneration and apoptosis induced by cytokines such as IL1β, IL-6, and IFN-α in human hippocampal neurogenesis and depression. These effects were mediated through LOX and CYP450-derived EPA/DHA metabolites [121]. Total erythrocyte n-3 PUFA concentrations are positively associated with cognitive function, particularly immediate recall, in older people with previous depression [122]. Similarly, higher levels of EPA and DHA erythrocytes were linked to improved depressive symptoms, especially following six months of high levels of EPA and DHA supplementation [123]. A recent review study revealed that n-3 PUFAs may represent a promising strategy for managing long-term COVID, as they can potentially alleviate chronic inflammation and restore tissue homeostasis, which can ultimately aid in the recovery from SARS-CoV-2 infection [124]. Similarly, while the primary management of mood disorders like depression in COPD involves the use of antidepressants, their tolerability is limited. However, as n-3 PUFAs are crucial in regulating inflammatory responses, they could be a promising alternative for managing mood disorders in chronic obstructive pulmonary disease (COPD) [125]. However, a further clinical trial is necessary to fully comprehend the potential advantages of n-3 PUFAs in managing these comorbidities and their long-term consequences.

It is noteworthy that safety considerations must be carefully evaluated and monitored in the development and implementation of personalized n-3 PUFA treatment approaches. While numerous findings suggest that n-3 PUFA treatment may be beneficial in improving neuropsychiatric and depressive symptoms in adults, it is crucial to acknowledge that n-3 PUFAs also impact depressive symptoms in vulnerable populations, including children, adolescents, and pregnant women [114,126,127]. In general, numerous clinical trials have demonstrated that n-3 PUFAs are generally well tolerated in diverse populations. For instance, EPA and DHA are well tolerated in hepatitis C infection patients; no effect of PUFA treatment on viral load, excessive bleeding, or liver function was observed [101]. n-3 PUFAs with a ratio of EPA/DHA of at least 1.5 (≥1.5) have been effective in treating mild-to-moderate depression in pregnant and postpartum women with minimal side effects. The trials included in the analysis did not show a significant difference in the incidence of gastrointestinal and neurologic adverse effects between the n-3 PUFAs and placebo groups [127]. In addition to youth with MDD, n-3 PUFA supplementation also showed its potential effects in the improvement of clinical symptoms in those with high inflammation or a low baseline n-3 index, particularly youth with attention deficit hyperactivity disorder (ADHD) and autism spectrum disorder (ASD), subthreshold psychotic states, and posttraumatic stress disorder (PTSD) [115,117,126]. Overall, n-3 PUFA supplementation provides no significant adverse effect and is a safe and effective therapy for these vulnerable groups [114,115,127]. However, developing personalized n-3 PUFA treatment requires a deep understanding of the molecular and genetic factors that contribute to disease susceptibility and progression. This is particularly important for patients with comorbid MDD and obesity, as well as metabolic dysregulation, where tailored n-3 PUFA treatments may be necessary to optimize outcomes and minimize potential adverse effects.

## 5. Omega-3 Polyunsaturated Fatty Acids (n-3 PUFAs) for Depression Treatment in Obese and Metabolically Dysregulated Patients

Selected key studies of n-3 PUFAs treatment of MDD, obesity, and metabolic dysregulation reviewed in this paper are summarized in Table 2.

Individuals with comorbid conditions such as depression, obesity, and metabolic dysregulation pose a challenge in terms of developing effective treatments due to the complex nature of these conditions. Bot et al. (2011) found no significant effect of EPA supplementation on BDNF levels in patients with both depression and diabetes, despite ongoing antidepressant use. However, the study authors noted that the baseline BDNF levels were relatively high, and ongoing antidepressant treatment throughout the study may have influenced the results but not depression severity [107].

Apart from diabetes, obesity is a frequently observed risk factor for metabolic syndrome and often coexists with depression. n-3 PUFAs at a dose of 1080 mg EPA and 720 mg DHA per day significantly reduced depression and body weight compared to placebo in patients with depression and comorbid obesity, but weight regain occurred in the follow-up period [108]. Nevertheless, several studies have indicated that n-3 PUFAs may have favorable effects on body composition and weight. For instance, a study revealed that the consumption of fish or fish oil led to reduced body weight, fat mass, and waist circumference in the n-3 PUFA group in comparison to the control group [128]. A reduction in energy intake, carbohydrates, and fat has been observed in overweight and obese females (25.8–39.9 kg/m^2^) with n-3 PUFA supplementation (2.8 g/day DHA) over 12 weeks; however, no significant effect has been implied on body weight [129]. Obese participants with a BMI of 30–40 kg/m^2^ given n-3 PUFA (180 mg EPA, 120 mg DHA) treatment for 4 weeks had reduced caloric intake, increased fullness, and non-significant changes in BMI and serum leptin levels, indicating potential effects on satiety and energy intake [130]. High-dose supplementation (4 g/d) of n-3 PUFAs has been shown to induce significant changes in adipose tissue, circulating fatty acids, and systemic inflammatory markers in individuals with obesity and insulin secretion, indicating the potential therapeutic effects of n-3 PUFAs on body weight [66].

N-3 PUFAs have been acknowledged for their potential therapeutic advantages in individuals with obesity, which is linked to long-term energy imbalance and adipocyte-derived hormones such as leptin and appetite sensations [131]. Depression is also associated with adiposity, and both conditions exacerbate each other. This vicious cycle is fueled by inflammation [61]. The response to EPA versus DHA may vary depending on baseline inflammatory biomarkers and adiponectin levels in those individuals who have high baseline inflammatory biomarkers [27]. Figure 1 shows an overview of n-3 PUFAs’ potential for managing obesity, metabolic dysregulation, and co-occurring depression [27,86,132,133,134]. A recent study revealed that high doses of EPA (4 g/d) may be more effective in treating resistant depression with severe inflammation, particularly in overweight individuals with high inflammation [26]. However, further research is needed to determine ideal dosages for EPA and DHA supplementation in specific subgroups with comorbidities of depression.

## 6. Future Perspective

The global health landscape faces significant challenges due to the co-occurrence of depression and obesity, as these conditions worsen each other’s effects and negatively impact patients’ quality of life and treatment outcomes. N-3 PUFA supplementation has emerged as a potential therapeutic strategy for addressing both depression and obesity, as it has been shown to have beneficial effects on inflammatory markers, body weight, and depressive symptoms. However, optimal dosing and duration of supplementation for individuals with comorbid depression, obesity, or metabolic dysregulation are still unclear. In the future, personalized and precision medicine approaches could play a crucial role in addressing this complex clinical challenge. Traditional anthropometric indices, such as body mass index (BMI), may not fully capture the complexity of obesity-related metabolic complications, and additional clinical parameters may be needed for accurate risk evaluation.

The traditional “one-size-fits-all” approach to treating depression is limited due to the heterogeneity of symptoms and treatment responses. Personalized treatment that considers the complexity of comorbidities and individual characteristics may improve outcomes for individuals with coexisting MDD and obesity. Therefore, a personalized medicine approach that integrates individualized adipose tissue health and metabolic parameters may offer more effective and tailored interventions. Further research is recommended to better understand how n-3 PUFAs work in obesity and MDD and to establish the most favorable dosage and duration of supplementation for people with comorbidities. Implementing personalized and precision medicine approaches in clinical practice could significantly improve treatment outcomes for individuals with comorbid MDD and obesity. Further investigation is warranted to improve the application of N-3 PUFA supplementation and sharpen up personalized medicine strategies, ultimately enhancing the efficacy of interventions for this complex patient population.

## Figures and Tables

**Figure 1 jpm-13-01003-f001:**
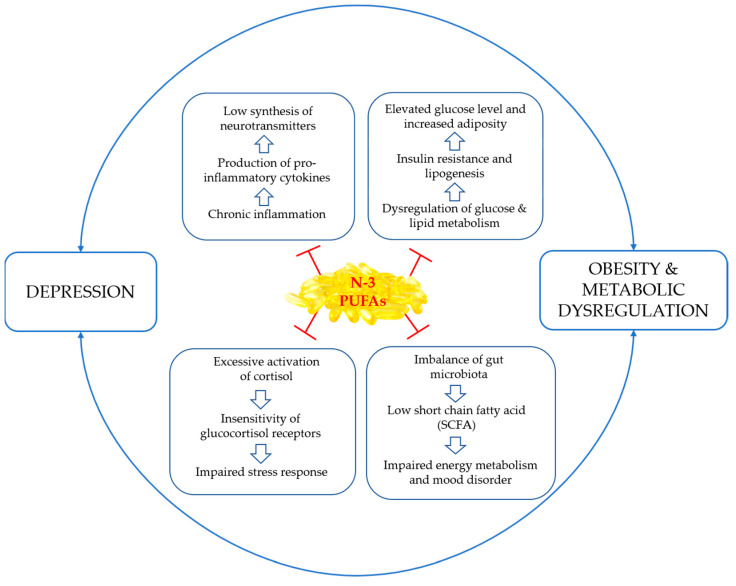
Summary of the potentially favorable effect of n-3 PUFAs in the treatment of depression comorbid with obesity and metabolic dysregulation.

**Table 1 jpm-13-01003-t001:** Personalized medicine of omega-3 fatty acids in depression based on two major clinical considerations.

Biological specificity		**Population**	**Study Design**	**Intervention**	**Main Findings**	**References**
	i. Depression associated with Inflammation					
	IFN-induced depression	*n* = 152, HCV patients	Double-blind, RCT	2-week treatment with EPA (3.5 g/day) (*n* = 50); DHA (1.75 g/day) (*n* = 51); placebo (*n* = 50)	EPA-treated, not DHA-treated significantly decreased the incidence of IFN-α-induced depression in HCV patients (10%, 28%, respectively, compared to 30% placebo, *p* = 0.037).	[101]
	Cardiovascular disease comorbidity	*n* = 59, CVD patients with comorbid MDD	RCT	2-week treatment with n-3 PUFAs (EPA 2.0 g/day and DHA 1.0 g/day) (*n* = 30); placebo (*n* = 29)	No significant differences between groups in HAMD and BDI-II total scores, but n-3 PUFAs significantly reduced HAMD cognition at week 8 (*p* < 0.05) and the HAMD core subscale in very severe MDD at week 12 (*p* < 0.05).	[104]
		*n* = 108, CHF patients with comorbid MDD	Double-blind, RCT	2-week treatment with n-3 PUFAs (2:1 EPA/DHA 2.0 g/day) (*n* = 36); EPA (2 g/day) (*n* = 36); placebo (*n* = 36)	No significant differences between groups in HAMD and BDI-II total scores. BDI-II cognitive depressive subscales were strongly associated with the high EPA group (*p* < 0.05).	[103]
		*n* = 92, CAD patients with comorbid MDD	Double-blind, RCT	2-week treatment with n-3 PUFAs (1.9 g/day) (*n* = 45); placebo (*n* = 47)	Plasma EPA and DHA levels increased (*p* < 0.01), but no significant differences between groups in HAMD (*p* = 0.20) and BDI-II (*p* = 0.50) total scores.	[105]
	Pain comorbidity	*n* = 46, Breast cancer survivor	Prospective, RCT	6-week diet intervention with high n-3 PUFAs (2040 mg, 12 ounces/week wild-caught Alaskan salmon) (*n* = 24) and low PUFAs (1020 mg, 6 ounces/week) (*n* = 15)	High n-3 PUFA group significantly decreased pain (*p* < 0.01), perceived stress (*p* < 0.05), sleep (*p* < 0.001), depression (*p* < 0.001), and fatigue (*p* < 0.01).	[106]
	Obesity and metabolic comorbidity	*n* = 25, Diabetes patients with MDD	Double-blind, RCT	12-week treatment with EPA (1 g/day) (*n* = 13), placebo (*n* = 12)	No significant impact on BDNF (*p* = 0.887) and no significant association between changes in BDNF levels and depression severity (*p* = 0.593).	[107]
		*n* = 45, Obese patient with MDD	Double-blind, RCT	12-week treatment with n-3 PUFAs (1.08 g/d EPA and 0.72 g/d DHA) (*n* = 24); placebo (*n* = 21)	Significantly reduced depression (*p* = 0.05) in n-3 treatment groups.	[108]
		*n* = 61, Obese patient with MDD	Double-blind, RCT	12-week treatment with EPA (1 g/d, *n* = 15; 2 g/d, *n* = 15; 4 g/day, *n* = 16); placebo (*n* = 15)	EPA 4 g/d produced a sustained effect on IDS-30 scores at both weeks 8 and 12. A potential dose-response relationship between EPA dose and change in IDS-C30 scores, but this was not statistically significant.	[26]
	ii. Depression associated with Neurodegeneration					
	Parkinson’s disease comorbidity	*n* = 29, Parkinson’s disease with MDD	Double-blind, RCT	12-week treatment with n-3 PUFAs (1200 mg/d) (*n* = 14); placebo (*n* = 15)	Significant decrease in MADRS and CGI-Depression scores but not in BDI in the n-3 PUFAs group.	[109]
	Alzheimer’s disease comorbidity	*n* = 26, Patients with MCI or AD	Double-blind, RCT	24-week treatment with n-3 PUFAs (1080 mg/d EPA and 720 mg/d DHA) (*n* = 20); placebo (*n* = 15)	No associations were found between the randomization group and ADAS-cog, MMSE, or HDRS scores.	[110]
		*n* = 204, Patient with AD	Double-blind, RCT	6-month treatment with n-3 PUFAs (0.6 g/d EPA and 1.7 g/d DHA) (*n* = 103); placebo (*n* = 103)	No overall n-3 PUFA effect on neuropsychiatric symptoms. Possible positive effects of MADRS in non-APOEv4 carriers (*p* = 0.005).	[111]
	Late-life depression	*n* = 18,353, Adults aged 50 years or older without depression	RCT	Median of 5.3 years of treatment with n-3 PUFAs (65 mg/d EPA and 375 mg/d DHA) (*n* = 9171), placebo (*n* = 9182)	No significant improvement in the n-3 PUFA group in the prevention of depression.	[112]
	iii. Physiological deficits					
	Perinatal depression	*n* = 59, Perinatal and postpartum women	Double-blind, RCT	8-week treatment with n-3 PUFAs (1.9 g/d) (*n* = 28), placebo (*n* = 23)	Both groups experienced significant decreases in EPDS and HAM-D scores (*p* < 0.0001) from baseline.	[113]
		*n* = 36, Perinatal women with MDD	Double-blind, RCT	8-week treatment with n-3 PUFAs (3.4 g/d) (*n* = 18), placebo (*n* = 18)	Significantly lower EDPS and BDI were observed in the n-3 PUFA group.	[22]
Safety consideration	Special populations					
	Children and adolescents with depression	*n* = 60, Children with DD or MADD	Double-blind, RCT	12-week treatment n-3 PUFAs (2.4 g/d) (*n* = 30), n-6 PUFAs (*n* = 30)	Significant reductions in CDI scores were observed in the n-3 PUFAs group and the DD subgroup compared to the n-6 PUFAs and MADD subgroup. No serious side effects were observed, except for increased defecation reported by one participant in the n-3 PUFA group.	[114]
	High-risk psychosis	*n* = 81, Patients with an ultra-high risk of psychotic disorder	Double-blind, RCT	12-week treatment with n-3 PUFAs (1.2 g/d) (*n* = 41), placebo (*n* = 40)	Significant improvement was observed in PANSS and GAF scores but not in MADRS scores. No significant adverse effects between the treatment groups.	[115]
		*n* = 50, Patients with schizophrenia-spectrum or bipolar disorders on medication	RCT	16-week n-3 PUFAs (740 mg/d EPA and 400 mg/d DHA) (*n* = 27), placebo (*n* = 26)	The n-3 PUFA group showed significant improvement in BPRS scores compared to placebo among a subgroup of patients (*n* = 23) who did not receive lorazepam. Lower rates of confusion, anxiety, depression, irritability, and tiredness/fatigue in the n-3 PUFAs group as compared to those on placebo.	[116]
		*n* = 110, Injured patients with PTSD	Double-blind, RCT	12-week n-3 PUFAs (1470 mg/d DHA and 147 mg/d EPA) (*n* = 53), placebo (*n* = 57)	Serum BDNF and pro-BDNF changes at week 12 were linked to depression severity, but DHA had no specific effect on these levels. Adverse events, including loose stool and constipation, were reported, but there were no significant differences between the two groups.	[117]
	Pregnant women with MDD	*n* = 36, Perinatal women with MDD	Double-blind, RCT	8-week treatment with n-3 PUFAs (3.4 g/d) (*n* = 18), placebo (*n* = 18)	Well tolerated, and there were no adverse effects on the subjects or newborns.	[22]

Abbreviation: RCT, randomized controlled trial; EPA, eicosapentaenoic acid; DHA, docosahexaenoic acid; MDD, major depressive disorder; HCV, hepatitis C viral infection; CVD, cardiovascular disease; CAD, coronary artery disease; CHF, chronic heart failure; HAMD, Hamilton Depression Rating Scale; BDI, Beck’s Depression Inventory; DD, depressive disorder; MADD, mixed anxiety and depressive disorder; GAF, global assessment of functioning; MADRS, Montgomery-Åsberg Depression Rating Scale; PANSS, positive and negative syndrome scale; BPRS, brief psychiatric rating scale; BDNF, brain-derived neurotrophic factor, CES-D, Center for Epidemiologic Studies Depression Scale, PTSD, post-traumatic stress disorder, EDPS, Edinburgh Postnatal Depression Scale.

**Table 2 jpm-13-01003-t002:** Key characteristics of preliminary studies of n-3 PUFAs treatment of MDD, obesity, and metabolic dysregulation.

References	Sample	Study Design	Intervention	Outcome Scale	Main Findings	Notes and Limitations
Bot et al. (2011) [107]	*n* = 25, Women with MDD and diabetes on antidepressants	Double-blind, RCT	12-week treatment with EPA (1 g/day) (*n* = 13), placebo (*n* = 12)	MADRS, Serum BDNF	No significant impact on BDNF (*p* = 0.887) and no significant association between changes in BDNF levels and depression severity (*p* = 0.593).	First clinical study to examine the effects of n-3 PUFAs on BDNF. Small sample size. One patient reported an allergic reaction and discontinued using EPA; no other severe adverse events were reported.
Keshavarz et al. (2018) [108]	*n* = 45, Women with depression and comorbid obesity (BMI ≥ 25 kg/m^2^) without antidepressants	Double-blind, RCT,	12-week treatment with n-3 PUFAs (EPA 1.08 g/day and DHA 0.72 g/day) (*n* = 24); placebo (*n* = 21)	Body weight, height, BMI, waist and hip circumferences, total body fat, muscle percentage, BDI, food craving questionnaire, appetite, and food abstinence visual Analogue scales	Reduced depression (*p* = 0.05) and body weight (*p* = 0.049) in n-3 treatment groups	Weight regains after a one-month follow-up. Side effects include nausea, skin rash, hemorrhagia, and increased appetite, which were reported in both groups.
Mischoulon et al. (2022) [26]	*n* = 61, Patient with MDD comorbid overweight/obese (BMI ≥ 25 kg/m^2^), without antidepressants	Double-blind, RCT	12-week treatment with EPA (1 g/d, *n* = 15; 2 g/d, *n* = 15; 4 g/day, *n* = 16); placebo (*n* = 15)	IL-6, LPS-stimulated TNF level, plasma hs-CRP level, IDS-C30	EPA 4 g/d produced a sustained effect on IDS-30 scores at both weeks 8 and 12. A potential dose-response relationship between EPA dose and change in IDS-C30 scores, but this was not statistically significant.	1st dose-finding trial of EPA in MDD to focus on inflammatory biomarkers as a primary outcome among overweight/obese subjects with elevated hs-CRP

Abbreviations: MDD, major depression disorder; BMI, body mass index; EPA, eicosapentaenoic acid; DHA, docosahexaenoic acid; MADRS, Montgomery-Asberg Depression Rating Scale; BNDF, brain-derived neurotropic factor; BDI, Beck Depression Inventory; IL-6, interleukin-6; LPS-stimulated TNF level, lipopolysaccharide-stimulated tumor necrosis factor; hs-CRP, high-sensitivity C-reactive protein; IDS-C30, inventory of depressive symptomatology, clinical-rated version.

## Data Availability

Not applicable.

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
