# Peer review of "Personalized Medicine of Omega-3 Fatty Acids in Depression Treatment in Obese and Metabolically Dysregulated Patients"

_jpm, 2023, doi:10.3390/jpm13061003_

Round 1

Reviewer 1 Report

The manuscript is well written and organized

Perhaps adding a figure to illustrate the mechanism of Omega-3 polyunsaturated fatty acids (n-3 PUFAs)  and its rationale is required

This is a nice piece of work

Reviewer 2 Report

The authors performed a review about the Personalized medicine of omega-3 fatty acids in depression treatment in obese and metabolically dysregulated patients.

Overall, this concise review is well written and provides important evidence for the emerging role of low-grade inflammation and metabolic abnormalities in the development of depression in which could enable the future development of personalized and effective treatment strategies for MDD patients with metabolic dysregulation.

I would recommend the manuscript be accepted with only minor grammatical review required.

Only minor grammatical review required.
